# GUIDED EVOLUTIONARY STRATEGIES: ESCAPING THE CURSE OF DIMENSIONALITY IN RANDOM SEARCH

## ABSTRACT

Many applications in machine learning require optimizing a function whose true gradient is unknown, but where *surrogate* gradient information (directions that may be correlated with, but not necessarily identical to, the true gradient) is available instead. This arises when an approximate gradient is easier to compute than the full gradient (e.g. in meta-learning or unrolled optimization), or when a true gradient is intractable and is replaced with a surrogate (e.g. in certain reinforcement learning applications or training networks with discrete variables). We propose *Guided Evolutionary Strategies*, a method for optimally using surrogate gradient directions along with random search. We define a search distribution for evolutionary strategies that is elongated along a subspace spanned by the surrogate gradients. This allows us to estimate a descent direction which can then be passed to a first-order optimizer. We analytically and numerically characterize the tradeoffs that result from tuning how strongly the search distribution is stretched along the guiding subspace, and use this to derive a setting of the hyperparameters that works well across problems. Finally, we apply our method to example problems including truncated unrolled optimization and training neural networks with discrete variables, demonstrating improvement over both standard evolutionary strategies and first-order methods (that directly follow the surrogate gradient). We provide a demo of Guided ES at: `redacted URL`

## 1  INTRODUCTION

Optimization in machine learning often involves minimizing a cost function where the gradient of the cost with respect to model parameters is known. When gradient information is available, first-order methods such as gradient descent are popular due to their ease of implementation, memory efficiency, and convergence guarantees (Sra et al., 2012). When gradient information is not available, however, we turn to zeroth-order optimization methods, including random search methods such as evolutionary strategies (Rechenberg, 1973; Nesterov & Spokoiny, 2011; Salimans et al., 2017).

However, what if only partial gradient information is available? That is, what if one has access to *surrogate* gradients that are correlated with the true gradient, but may be biased in some unknown fashion? Naïvely, there are two extremal approaches to optimization with surrogate gradients. On one hand, you could ignore the surrogate gradient information entirely and perform zeroth-order optimization, using methods such as evolutionary strategies to estimate a descent direction. These methods exhibit poor convergence properties when the parameter dimension is large (Duchi et al., 2015). On the other hand, you could directly feed the surrogate gradients to a first-order optimization algorithm. However, bias in the surrogate gradients will interfere with optimizing the target problem (Tucker et al., 2017). Ideally, we would like a method that combines the complementary strengths of these two approaches: we would like to combine the unbiased descent direction estimated with evolutionary strategies with the low-variance estimate given by the surrogate gradient. In this work, we propose a method for doing this called guided evolutionary strategies (Guided ES).

The critical assumption underlying Guided ES is that we have access to surrogate gradient information, but not the true gradient. This scenario arises in a wide variety of machine learning problems, which typically fall into two categories: cases where the true gradient is unknown or not defined, and cases where the true gradient is hard or expensive to compute. Examples of the former include: models with discrete stochastic variables (where straight through estimators (Bengio et al.,

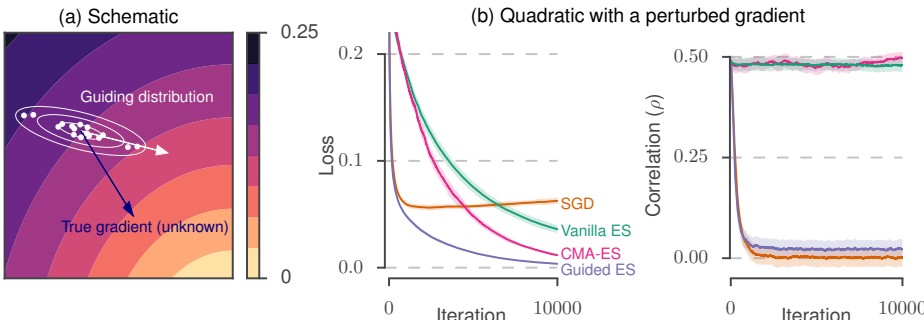

Figure 1: (a) Schematic of guided evolutionary strategies. We perform a random search using a distribution (white contours) elongated along a subspace (white arrow) which we are given instead of the true gradient (blue arrow). (b) Comparison of different algorithms on a quadratic loss, where a bias is explicitly added to the gradient to mimic situations where the true gradient is unknown. The loss (left) and correlation between surrogate and true gradient (right) are shown during optimization. See §4.1 for experimental details.

2013; van den Oord et al., 2017) or Concrete/Gumble-Softmax methods (Maddison et al., 2016; Jang et al., 2016) are commonly used) and learned models in reinforcement learning (e.g. for Q functions (Watkins & Dayan, 1992; Mnih et al., 2013; 2015; Lillicrap et al., 2015) or value estimation (Mnih et al., 2016)). For the latter, examples include optimization using truncated backprop through time (Rumelhart et al., 1985; Williams & Peng, 1990; Wu et al., 2018). Surrogate gradients also arise in situations where the gradients are explicitly modified during training, as in feedback alignment (Lillicrap et al., 2014) and related methods (Nøkland, 2016; Gilmer et al., 2017).

The key idea in Guided ES is to keep track of a low-dimensional subspace, defined by the recent history of surrogate gradients during optimization, which we call the *guiding subspace*. We then perform a finite difference random search (as in evolutionary strategies) preferentially within this subspace. By concentrating our search samples in a low-dimensional subspace where the true gradient has non-negative support, we dramatically reduce the variance of the search direction.

Our contributions in this work are:

- a new method for combining surrogate gradient information with random search,
- an analysis of the bias-variance tradeoff underlying the technique (§3.3),
- a scheme for choosing optimal hyperparameters for the method (§3.4), and
- applications to example problems (§4).

## 2    RELATED WORK

This work builds upon a random search method known as evolutionary strategies (Rechenberg, 1973; Nesterov & Spokoiny, 2011), or ES for short, which generates a descent direction via finite differences over random perturbations of parameters. ES has seen a resurgence in popularity in recent years (Salimans et al., 2017; Mania et al., 2018). Our method can primarily be thought of as a modification to ES where we augment the search distribution using surrogate gradients.

Extensions of ES that modify the search distribution use natural gradient updates in the search distribution (Wierstra et al., 2008) or construct non-Gaussian search distributions (Glasmachers et al., 2010). The idea of using gradients in concert with evolutionary algorithms was proposed by Lehman et al. (2017b), who use gradients of a network with respect to its *inputs* (as opposed to parameters) to augment ES. Other methods for adapting the search distribution include covariance matrix adaptation ES (CMA-ES) (Hansen, 2016), which uses the recent history of descent steps to adapt the distribution over parameters, or variational optimization (Staines & Barber, 2012), which optimizes the parameters of a probability distribution over model weights. Guided ES, by contrast, adapts the search distribution using surrogate gradient information. In addition, we never need to work with or compute a full $n \times n$ covariance matrix.

## 3 GUIDED EVOLUTIONARY STRATEGIES

### 3.1 VANILLA ES

We wish to minimize a function $f(x)$ over a parameter space in $n$-dimensions ($x \in \mathbb{R}^n$), where $\nabla f$ is either unavailable or uninformative. A popular approach is to estimate a descent direction with stochastic finite differences (commonly referred to as evolutionary strategies (Rechenberg, 1973) or random search (Rastrigin, 1963)). Here, we use antithetic sampling (Owen, 2013) (using a pair of function evaluations at $x + \epsilon$ and $x - \epsilon$) to reduce variance. This estimator is defined as:

$$g = \frac{\beta}{2\sigma^2 P} \sum_{i=1}^{P} \epsilon_i \left( f(x + \epsilon_i) - f(x - \epsilon_i) \right), \tag{1}$$

where $\epsilon_i \sim \mathcal{N}(0, \sigma^2 I)$, and $P$ is the number of sample pairs. We will set $P$ to one for all experiments, and when analyzing optimal hyperparameters. The overall scale of the estimate ($\beta$) and variance of the perturbations ($\sigma^2$) are constants, to be chosen as hyperparameters. This estimate solely relies on computing $2P$ function evaluations. However, it tends to have high variance, thus requiring a large number of samples to be practical, and scales poorly with the dimension $n$. We refer to this estimator as vanilla evolutionary strategies (or vanilla ES) in subsequent sections.

### 3.2 GUIDED SEARCH

Even when we do not have access to $\nabla f$, we frequently have additional information about $f$, either from prior knowledge or gleaned from previous iterates during optimization. To formalize this, we assume we are given a set of vectors which may correspond to biased or corrupted gradients. That is, these vectors are correlated (but need not be perfectly aligned) with the true gradient. If we are given a single vector or surrogate gradient for a given parameter iterate, we can generate a subspace by keeping track of the previous $k$ surrogate gradients encountered during optimization. We use $U$ to denote an $n \times k$ orthonormal basis for the subspace spanned by these vectors (i.e., $U^T U = I_k$).

We leverage this information by changing the distribution of $\epsilon_i$ in eq. (1) to $\mathcal{N}(0, \sigma^2 \Sigma)$ with

$$\Sigma = \frac{\alpha}{n} I_n + \frac{1 - \alpha}{k} U U^T,$$

where $k$ and $n$ are the subspace and parameter dimensions, respectively, and $\alpha$ is a hyperparameter that trades off variance between the full parameter space and the subspace. Setting $\alpha = 1$ recovers the vanilla ES estimator (and ignores the guiding subspace), but as we show choosing $\alpha < 1$ can result in significantly improved performance. The other hyperparameter is the scale $\beta$ in (1), which controls the size of the estimated descent direction. The parameter $\sigma^2$ controls the overall scale of the variance, and will drop out of the analysis of the bias and variance below, due to the $\frac{1}{\sigma^2}$ factor in (1). In practice, if $f(x)$ is stochastic, then increasing $\sigma^2$ will dampen noise in the gradient estimate, while decreasing $\sigma^2$ reduces the error induced by third and higher-order terms in the Taylor expansion of $f$ below. For an exploration of the effects of $\sigma^2$ in ES, see Lehman et al. (2017a).

Samples of $\epsilon_i$ can be generated efficiently as $\epsilon_i = \sigma \sqrt{\frac{\alpha}{n}} \epsilon + \sigma \sqrt{\frac{1-\alpha}{k}} U \epsilon'$ where $\epsilon \sim N(0, I_n)$ and $\epsilon' \sim N(0, I_k)$. Our estimator requires $2P$ function evaluations in addition to the cost of computing the surrogate gradient. Furthermore, it may be possible to parallelize the forward pass computations.

Figure 1a depicts the geometry underlying our method. Instead of the true gradient (blue arrow), we are given a surrogate gradient (white arrow) which is correlated with the true gradient. We use this to form a guiding distribution (denoted with white contours) and use this to draw samples (white dots) which we use as part of a random search procedure. (Figure 1b demonstrates the performance of the method on a toy problem, and is discussed in §4.1.)

For the purposes of analysis, suppose $\nabla f$ exists. We can approximate the function in the local neighborhood of $x$ using a second order Taylor approximation: $f(x + \epsilon) \approx f(x) + \epsilon^T \nabla f(x) + \frac{1}{2} \epsilon^T \nabla^2 f(x) \epsilon$. For the remainder of §3, we take this second order Taylor expansion to be exact. By substituting this expression into (1), we see that our estimate $g$ is equal to

$$g = \frac{\beta}{\sigma^2 P} \sum_{i=1}^{P} \left( \epsilon_i \epsilon_i^T \right) \nabla f(x). \tag{2}$$

Note that even terms in the Taylor expansion cancel out in the expression for $g$ due to antithetic sampling. The computational and memory costs of using Guided ES to compute parameter updates, compared to standard (vanilla) ES and gradient descent, are outlined in Appendix D.

### 3.3 TRADEOFF BETWEEN VARIANCE AND SAFE BIAS

As we have alluded to, there is a bias-variance tradeoff lurking within our estimate $g$. In particular, by emphasizing the search in the full space (i.e., choosing $\alpha$ close to 1), we reduce the bias in our estimate at the cost of increased variance. Emphasizing the search along the guiding subspace (i.e., choosing $\alpha$ close to 0) will induce a bias in exchange for a potentially large reduction in variance, especially if the subspace dimension $k$ is small relative to the parameter dimension $n$. Below, we analytically and numerically characterize this tradeoff.

Importantly, regardless of the choice of $\alpha$ and $\beta$, the Guided ES estimator always provides a descent direction in expectation. The mean of the estimator in eq. (2) is $\mathbb{E}[g] = \beta \Sigma \nabla f(x)$ corresponds to the gradient multiplied by a positive semi-definite (PSD) matrix, thus the update $(-\mathbb{E}[g])$ remains a descent direction. This desirable property ensures that $\alpha$ trades off variance for "safe" bias. That is, the bias will never produce an *ascent* direction when we are trying to *minimize* $f$.

The alignment between the $k$-dimensional orthonormal guiding subspace ($U$) and the true gradient ($\nabla f(x)$) will be a key quantity for understanding the bias-variance tradeoff. We characterize this alignment using a $k$-dimensional vector of uncentered correlation coefficients $\rho$, whose elements are the correlation between the gradient and every column of $U$. That is, $\rho_i = \frac{\nabla f(x)^T U_{\cdot i}}{\|\nabla f(x)\|}$. This correlation $\|\rho\|_2$ varies between zero (if the gradient is orthogonal to the subspace) and one (if the gradient is full contained in the subspace).

We can evaluate the squared norm of the bias of our estimate $g$ as

$$\| \operatorname{Bias}(g)\|_2^2 = (\mathbb{E}[g] - \nabla f(x))^T (\mathbb{E}[g] - \nabla f(x))$$
$$= \nabla f(x)^T (\beta \Sigma - I)^2 \nabla f(x). \tag{3}$$

We additionally define the *normalized* squared bias, $\tilde{b}$, as the squared norm of the bias divided by the squared norm of the true gradient (this quantity is independent of the overall scale of the gradient). Plugging in our estimate for $g$ from eq. (2) yields the following expression for the normalized squared bias (see Appendix A.1 for derivation):

$$\tilde{b} = \left(\beta \frac{\alpha}{n} - 1\right)^2 + \left(\beta^2 \frac{(1-\alpha)^2}{k^2} + 2\beta \frac{(1-\alpha)}{k} \left(\beta \frac{\alpha}{n} - 1\right)\right) \|\rho\|_2^2 \tag{4}$$

where again $\beta$ is a scale factor and $\alpha$ is part of the parameterization of the covariance matrix that trades off variance in the full parameter space for variance in the guiding subspace ($\Sigma = \frac{\alpha}{n}I + \frac{(1-\alpha)}{k}UU^T$). We see that the normalized squared bias consists of two terms: the first is a contribution from the search in the full space and is thus independent of $\rho$, whereas the second depends on the squared norm of the uncentered correlation, $\|\rho\|_2^2$.

In addition to the bias, we are also interested in the variance of our estimate. We use total variance (i.e., $\operatorname{tr}(\operatorname{Var}(g))$) to quantify the variance of our estimator

$$\text{total variance} \equiv \operatorname{tr}\left(\operatorname{Var}(g)\right) = \operatorname{tr}\left(\mathbb{E}[gg^T] - \mathbb{E}[g]\mathbb{E}[g]^T\right) = \mathbb{E}[g^T g] - \mathbb{E}[g]^T \mathbb{E}[g]$$
$$= \beta^2 \nabla f(x)^T \mathbb{E}[\epsilon \epsilon^T \epsilon \epsilon^T] \nabla f(x) - \beta^2 \nabla f(x)^T \Sigma^T \Sigma \nabla f(x)$$
$$= \nabla f(x)^T \left(\beta^2 \Sigma + \beta^2 \Sigma^2\right) \nabla f(x),$$

using an identity for the fourth moment of a Gaussian (see Appendix A.2) and the fact that the trace is linear and invariant under cyclic permutations.

We are interested in the normalized variance, $\tilde{v}$, which we define as the quantity above divided by the squared norm of the gradient. Plugging in our estimate $g$ yields the following expression for the normalized variance (see Appendix A.2):

$$\tilde{v} = \beta^2 \left(\frac{\alpha^2}{n^2} + \frac{\alpha}{n}\right) + \beta^2 \left(\frac{(1-\alpha)^2}{k^2} + 2\frac{\alpha(1-\alpha)}{kn} + \frac{(1-\alpha)}{k}\right) \|\rho\|_2^2. \tag{5}$$

Equations (4) and (5) quantify the bias and variance of our estimate as a function of the subspace and parameter dimensions ($k$ and $n$), the parameters of the distribution ($\alpha$ and $\beta$), and the correlation $\|\rho\|_2$. Note that for simplicity we have set the number of pairs of function evaluations, $P$, to one. As $P$ increases, the variance will decrease linearly, at the cost of extra function evaluations.

Figure 2 explores the tradeoff between normalized bias and variance for different settings of the relevant hyperparameters ($\alpha$ and $\beta$) for example values of $\|\rho\|_2 = 0.23$, $k = 3$, and $n = 100$. Figure 2c shows the sum of the normalized bias plus variance, the global minimum of which (blue star) can be used to choose optimal values for the hyperparameters, discussed in the next section.

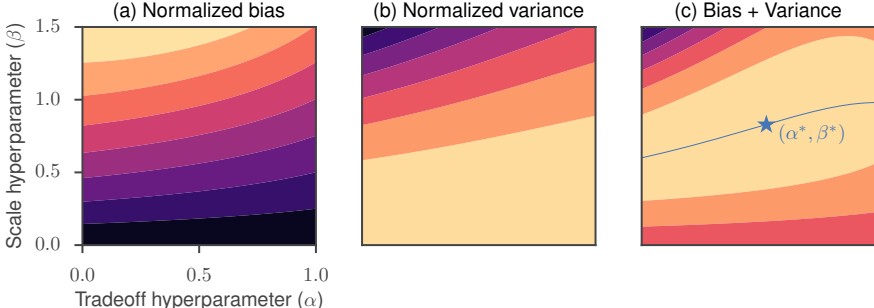

Figure 2: Exploring the tradeoff between variance and safe bias in Guided ES. Contour plots of normalized bias $\tilde{b}$ (a), normalized variance $\tilde{v}$ (b), and the sum of both (c) are shown as a function of the tradeoff ($\alpha$) and scale ($\beta$) hyperparameters, for a fixed $\|\rho\|_2 = 0.23$. For these plots, the subspace dimension was set to $k = 3$ and the parameter dimension was set to $n = 100$. The blue line in (c) denotes the optimal $\beta$ for every value of $\alpha$, and the star denotes the global optimum.

### 3.4 CHOOSING OPTIMAL HYPERPARAMETERS BY MINIMIZING ERROR IN THE ESTIMATE

The expressions for the normalized bias and variance depend on the subspace and parameter dimensions ($k$ and $n$, respectively), the hyperparameters of the guiding distribution ($\alpha$ and $\beta$) and the uncentered correlation between the true gradient and the subspace ($\|\rho\|_2$). All of these quantities except for the correlation $\|\rho\|_2$ are known or defined in advance.

To choose optimal hyperparameters, we minimize the sum of the normalized bias and variance, (equivalent to the expected normalized square error in the gradient estimate, $\tilde{b} + \tilde{v} = \frac{\mathbb{E}\left[\|g - \nabla f(x)\|_2^2\right]}{\|\nabla f(x)\|_2^2}$). This objective becomes:

$$\tilde{b} + \tilde{v} = \tag{6}$$
$$\left[2\beta^2 \frac{\alpha^2}{n^2} + (\beta^2 - 2\beta)\frac{\alpha}{n} + 1\right] + \left[2\beta^2 \frac{(1-\alpha)^2}{k^2} + 4\beta^2 \frac{\alpha(1-\alpha)}{kn} + (\beta^2 - 2\beta)\frac{(1-\alpha)}{k}\right]\|\rho\|_2^2,$$

subject to the feasibility constraints $\beta \geq 0$ and $0 \leq \alpha \leq 1$.

As further motivation for this hyperparameter objective, in the simple case that $f(x) = \frac{1}{2}\|x\|_2^2$ then minimizing eq. (6) also results in the hyperparameters that cause SGD to most rapidly descend $f(x)$. See Appendix C for a derivation of this relationship.

We can solve for the optimal tradeoff ($\alpha^*$) and scale ($\beta^*$) hyperparameters as a function of $\|\rho\|_2$, $k$, and $n$. Figure 3a shows the optimal value for the tradeoff hyperparameter ($\alpha^*$) in the 2D plane spanned by the correlation ($\|\rho\|_2$) and ratio of the subspace dimension to the parameter dimension $\frac{k}{n}$. Remarkably, we see that for large regions of the ($\|\rho\|_2, \frac{k}{n}$) plane, the optimal value for $\alpha$ is either 0 or 1. In the upper left (blue) region, the subspace is of high quality (highly correlated with the true gradient) and small relative to the full space, so the optimal solution is to place all of the weight in the subspace, setting $\alpha$ to zero (therefore $\Sigma \propto UU^T$). In the bottom right (orange) region, we have the opposite scenario, where the subspace is large and low-quality, thus the optimal solution is to place all of the weight in the full space, setting $\alpha$ to one (equivalent to vanilla ES, $\Sigma \propto I$). The strip in the middle is an intermediate regime where the optimal $\alpha$ is between 0 and 1.

We can also derive an expression for *when* this transition in optimal hyperparameters occurs. To do this, we use the reparameterization $\theta = \begin{pmatrix} \alpha\beta \\ (1-\alpha)\beta \end{pmatrix}$. This allows us to express the objective in (6) as a least squares problem $\frac{1}{2}\|A\theta - b\|_2^2$, subject to a non-negativity constraint ($\theta \succeq 0$), where $A$ and $b$ depend solely on the problem data $k$, $n$, and $\|\rho\|_2$ (see Appendix B.1 for details). In addition, $A$ is always a positive semi-definite matrix, so the reparameterized problem is convex. We are particularly interested in the point where the non-negativity constraint becomes tight. Formulating the Lagrange dual of this problem and solving for the KKT conditions allows us to identify this point using the complementary slackness conditions (Boyd & Vandenberghe, 2004). This yields the equations $\|\rho\|_2 = \sqrt{\frac{k+4}{n+4}}$ and $\|\rho\|_2 = \sqrt{\frac{k}{n}}$ (see Appendix B.2), which are shown in Figure 3a, and line up with the numerical solution. Figure 3b further demonstrates this tradeoff. For fixed $n = 100$, we plot four curves for $k$ ranging from 1 to 30. As $\|\rho\|_2$ increases, the optimal hyperparameters sweep out a curve from $\left(\alpha^* = 1, \beta^* = \frac{n}{n+2}\right)$ to $\left(\alpha^* = 0, \beta^* = \frac{k}{k+2}\right)$.

In practice, the correlation between the gradient and the guiding subspace is typically unknown. However, we find that ignoring $\|\rho\|_2$ and setting $\beta = 2$ and $\alpha = \frac{1}{2}$ works well (these are the values used for all experiments in this paper). A direction for future work would be to estimate the correlation $\|\rho\|_2$ online, and to use this to choose hyperparameters by minimizing eq. (6).

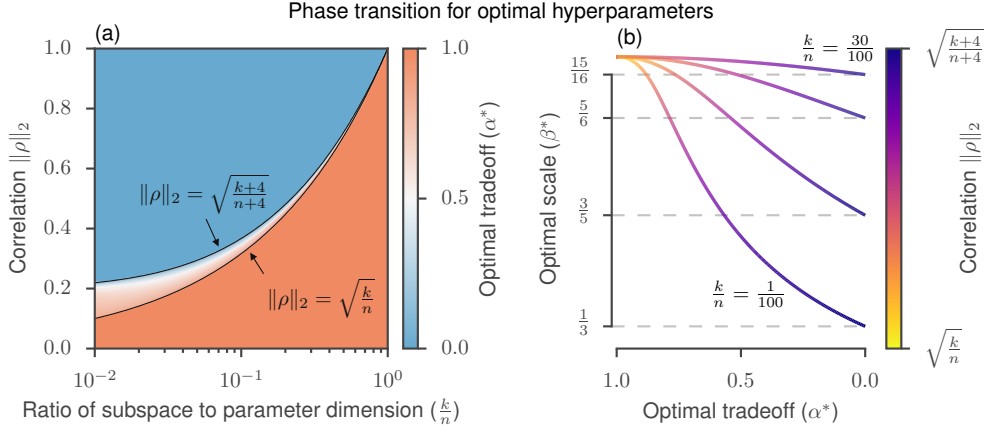

Figure 3: Choosing optimal hyperparameters. (a) Different regimes of optimal hyperparameters in the $(\frac{k}{n}, \|\rho\|_2)$ plane are shown as shaded regions. See §3.4 for details. (b) As $\|\rho\|_2$ increases, the optimal hyperparameters sweep out a curve from $\left(\alpha^* = 1, \beta^* = \frac{n}{n+2}\right)$ to $\left(\alpha^* = 0, \beta^* = \frac{k}{k+2}\right)$.

## 4 APPLICATIONS

### 4.1 QUADRATIC FUNCTION WITH A BIASED GRADIENT

We first test our method on a toy problem where we control the bias of the surrogate gradient explicitly. We generated random quadratic problems of the form $f(x) = \frac{1}{2}\|Ax - b\|_2^2$ where the entries of $A$ and $b$ were drawn independently from a standard normal distribution, but rather than allow the optimizers to use the true gradient, we (for illustrative purposes) added a random bias to generate surrogate gradients. Figure 1b compares the performance of stochastic gradient descent (SGD) with standard (vanilla) evolutionary strategies (ES), CMA-ES, and Guided ES. For this, and all of the results in this paper, we set the hyperparameters as $\beta = 2$ and $\alpha = \frac{1}{2}$, as described above.

We see that Guided ES proceeds in two phases: it initially quickly descends the loss as it follows the biased gradient, and then transitions into random search. Vanilla ES and CMA-ES, however, do not get to take advantage of the information available in the surrogate gradient, and converge more slowly. We see this also in the plot of the uncentered correlation ($\rho$) between the true gradient and the surrogate gradient in Figure 1c. Further experimental details are provided in Appendix E.1.

## 4.2 UNROLLED OPTIMIZATION

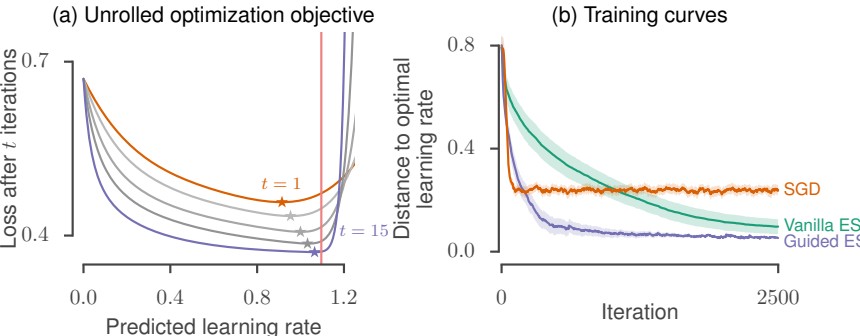

Figure 4: Unrolled optimization. (a) Bias in the loss landscape of unrolled optimization for small numbers of unrolled optimization steps ($t$). (b) Training curves (shown as distance from the optimum) for training a multi-layer perceptron to predict the optimal learning rate as a function of the eigenvalues of the function to optimize. See §4.2 for details.

Another application where surrogate gradients are available is in *unrolled* optimization. Unrolled optimization refers to taking derivatives through an optimization process. For example, this approach has been used to optimize hyperparameters (Domke, 2012; Maclaurin et al., 2015; Baydin et al., 2017), to stabilize training (Metz et al., 2016), and even to train neural networks to act as optimizers (Andrychowicz et al., 2016; Wichrowska et al., 2017; Li & Malik, 2017; Lv et al., 2017). Taking derivatives through optimization with a large number of steps is costly, so a common approach is to instead choose a small number of unrolled steps, and use that as a target for training. However, Wu et al. (2018) recently showed that this approach yields biased gradients.

To demonstrate the utility of Guided ES here, we trained multi-layer perceptrons (MLP) to predict the learning rate for a target problem, using as input the eigenvalues of the Hessian at the current iterate. Figure 4a shows the bias induced by unrolled optimization, as the number of optimization steps ranges from one iteration (orange) to 15 (blue). We compute the surrogate gradient of the parameters in the MLP using the loss after one SGD step. Figure 4b, we show the absolute value of the difference between the optimal learning rate and the MLP prediction for different optimization algorithms. Further experimental details are provided in Appendix E.2.

## 4.3 SYNTHESIZING GRADIENTS FOR A GUIDING SUBSPACE

Next, we explore using Guided ES in the scenario where the surrogate gradient is not provided, but instead we train a model to generate surrogate gradients (we call these synthetic gradients). In

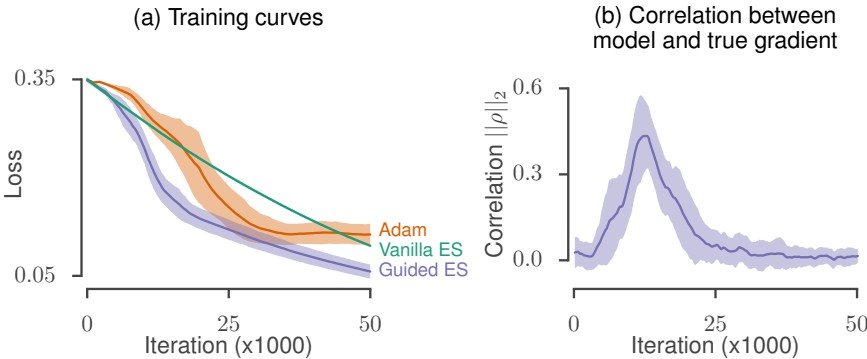

Figure 5: Synthetic gradients serving as the guiding subspace for Guided ES. (a) Loss curves when using synthetic gradients to minimize a target quadratic problem. (b) Correlation between the synthetic update direction and the true gradient during optimization for Guided ES.

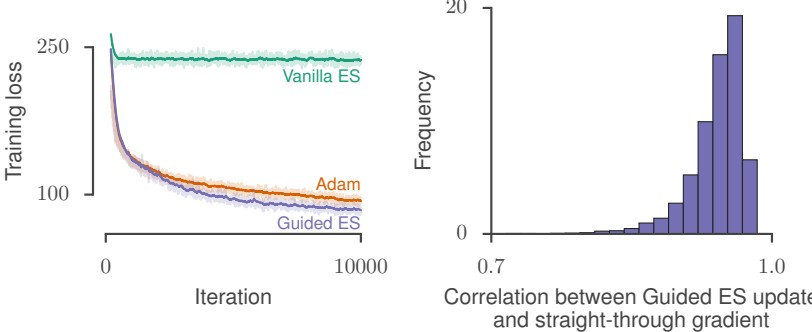

Figure 6: Training a VQ-VAE. (a) Guided ES (using the straight-through estimator as the surrogate gradient) achieves lower training loss than Adam. (b) Histogram of the correlation between the Guided ES update and the straight-through gradient during training.

real-world applications, training a model to produce synthetic gradients is the basis of model-based and actor-critic methods in RL (Lillicrap et al., 2015; Heess et al., 2015) and has been applied to decouple training across neural network layers (Jaderberg et al., 2016) and to generate policy gradients (Houthooft et al., 2018). A key challenge with such an approach is that early in training, the model generating the synthetic gradients is untrained, and thus will produce biased gradients. In general, it is unclear during training when following these synthetic gradients will be beneficial.

We define a parametric model, $M(x; \theta)$ (an MLP), which provides synthetic gradients for the target problem $f$. The target model $M(\cdot)$ is trained online to minimize mean squared error against evaluations of $f(x)$. Figure 5 compares vanilla ES, Guided ES, and the Adam optimizer (Kingma & Ba, 2014). We show training curves for these methods in Figure 5a, and the correlation between the synthetic gradient and true gradients for Guided ES in Figure 5b. Despite the fact that the quality of the synthetic gradients varies wildly during optimization, Guided ES consistently makes progress on the target problem. Further experimental details are provided in Appendix E.3.

### 4.4 NEURAL NETWORKS WITH DISCRETE LATENT VARIABLES

Finally, we applied Guided ES to train neural networks with discrete variables. Specifically, we trained autoencoders with a discrete latent codebook as in the VQ-VAE (van den Oord et al., 2017) on MNIST. The encoder and decoder were fully connected networks with two hidden layers. We use the straight-through estimator (Bengio et al., 2013) taken through the discretization step as the surrogate gradient. For Guided ES, we computed the Guided ES update only for the encoder weights, as those are the only parameters with biased gradients (due to the straight-through estimator)–the other weights in the network were trained directly with Adam. Figure 6a shows the training loss using Adam, standard (vanilla) ES, and Guided ES (note that vanilla ES does not make progress on this timescale due to the large number of parameters ($n = 152912$)). We achieve a small improvement, likely due to the biased straight-through gradient estimator leading to suboptimal encoder weights. The correlation between the Guided ES update step and the straight-through gradient (Figure 6b) can be thought of as a metric for the quality of the surrogate gradient (which is fairly high for this problem). Overall, this demonstrates that we can use Guided ES and first-order methods together, applying the Guided ES update only to the parameters that have surrogate gradients (and using first-order methods for the parameters that have unbiased gradients). Further experimental details are provided in Appendix E.4.

## 5 DISCUSSION

We have introduced guided evolutionary strategies (Guided ES), an optimization algorithm which combines the benefits of first-order methods and random search, when we have access to surrogate gradients that are correlated with the true gradient. We analyzed the bias-variance tradeoff inherent in our method analytically, and demonstrated the generality of the technique by applying it to unrolled optimization, synthetic gradients, and training neural networks with discrete variables.

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

## APPENDIX

## A DERIVATION OF THE BIAS AND VARIANCE OF THE GUIDED ES UPDATE

### A.1 BIAS

The squared bias norm is defined as:

$$\|\mathbb{E}[g] - \nabla f(x)\|_2^2 = \nabla f(x)^T \left(\beta\Sigma - I\right)^2 \nabla f(x),$$

where $\epsilon \sim \mathcal{N}(0, \Sigma)$ and the covariance is given by: $\Sigma = \frac{\alpha}{n}I + \frac{1-\alpha}{k}UU^T$. This expression reduces to (recall that $U$ is orthonormal, so $U^T U = I$):

$$\|\text{Bias}\|_2^2 = \|\nabla f(x)\|_2^2 \left[\beta^2\frac{\alpha^2}{n^2} - 2\beta\frac{\alpha}{n} + 1 + \left(\beta^2\frac{(1-\alpha)^2}{k^2} + 2\beta^2\frac{\alpha(1-\alpha)}{kn} - 2\beta\frac{1-\alpha}{k}\right)\|\rho\|_2^2\right]$$

$$= \|\nabla f(x)\|_2^2 \left[\left(\beta\frac{\alpha}{n} - 1\right)^2 + \left(\beta^2\frac{(1-\alpha)^2}{k^2} + 2\beta\frac{1-\alpha}{k}\left(\beta\frac{\alpha}{n} - 1\right)\right)\|\rho\|_2^2\right]$$

Dividing by the norm of the gradient ($\|\nabla f(x)\|_2^2$) yields the expression for the normalized bias (eq. (4) in the main text).

### A.2 VARIANCE

First, we state a useful identity. Suppose $\epsilon \sim \mathcal{N}(0, \Sigma)$, then

$$\mathbb{E}[\epsilon\epsilon^T\epsilon\epsilon^T] = \text{tr}(\Sigma)\Sigma + 2\Sigma^2.$$

We can see this by observing that the $(i, k)$ entry of $\mathbb{E}[\epsilon\epsilon^T\epsilon\epsilon^T] = \mathbb{E}[(\epsilon^T\epsilon)\epsilon\epsilon^T]$ is

$$\mathbb{E}\left[\sum_j \epsilon_i\epsilon_j^2\epsilon_k\right] = \sum_j \mathbb{E}\left[\epsilon_i\epsilon_j^2\epsilon_k\right]$$

$$= \sum_j \mathbb{E}\left[\epsilon_j^2\right]\mathbb{E}\left[\epsilon_i\epsilon_k\right] + 2\sum_j \mathbb{E}\left[\epsilon_i\epsilon_j\right]\mathbb{E}\left[\epsilon_j\epsilon_k\right],$$

by Isserlis' theorem, and then we recover the identity by rewriting the terms in matrix notation.

The total variance is given by:

$$\text{total variance} = \text{tr}(\text{Var}(g)) = \beta^2\nabla f(x)^T\mathbb{E}[\epsilon\epsilon^T\epsilon\epsilon^T]\nabla f(x) - \mathbb{E}[g]^T\mathbb{E}[g]$$

Using the identity above, we can express the total variance as:

$$\text{total variance} = \beta^2\nabla f(x)^T\left(\text{tr}(\Sigma)\Sigma + 2\Sigma^2\right)\nabla f(x) - \beta^2\nabla f(x)^T\Sigma^2\nabla f(x)$$

$$= \beta^2\nabla f(x)^T\left(\text{tr}(\Sigma)\Sigma + \Sigma^2\right)\nabla f(x)$$

Since the trace of the covariance matrix $\Sigma$ is 1, we can expand the quantity $\text{tr}(\Sigma)\Sigma + \Sigma^2$ as:

$$
\text{tr}(\Sigma)\Sigma + \Sigma^2 = \Sigma + \Sigma^2
$$
$$
= \left[\frac{\alpha^2}{n^2} + \frac{\alpha}{n}\right] I + \left[\frac{(1-\alpha)^2}{k^2} + 2\frac{\alpha(1-\alpha)}{kn} + \frac{1-\alpha}{k}\right] UU^T
$$

Thus the expression for the total variance reduces to:

$$
\text{total variance} = \|\nabla f(x)\|_2^2 \beta^2 \left(\frac{\alpha^2}{n^2} + \frac{\alpha}{n} + \left[\frac{(1-\alpha)^2}{k^2} + 2\frac{\alpha(1-\alpha)}{kn} + \frac{1-\alpha}{k}\right] \|\rho\|_2^2\right),
$$

and dividing by the norm of the gradient yields the expression for the normalized variance (eq. (5) in the main text).

## B  OPTIMAL HYPERPARAMETERS

### B.1  REPARAMETERIZATION

We wish to minimize the sum of the normalized bias and variance, eq. (6) in the main text. First, we use a reparameterization by using the substitution $\theta_1 = \alpha\beta$ and $\theta_2 = (1-\alpha)\beta$. This substitution yields:

$$
\tilde{b} + \tilde{v} = \left[2\frac{\theta_1^2}{n^2} + (\theta_0 + \theta_1 - 2)\frac{\theta_0}{n} + 1\right] + \left[2\frac{\theta_2^2}{k^2} + 4\frac{\theta_0\theta_1}{kn} + (\theta_0 + \theta_1 - 2)\frac{\theta_1}{k}\right] \|\rho\|_2^2,
$$

which is quadratic in $\theta$. Therefore, we can rewrite the problem as: $\tilde{b} + \tilde{v} = \frac{1}{2}\|A\theta - b\|_2^2$, where $A$ and $b$ are given by:

$$
A = \begin{pmatrix} \frac{2}{n^2} + \frac{1}{n} & \frac{1}{2}\left(\frac{4\|\rho\|_2^2}{kn} + \frac{\|\rho\|_2^2}{k} + \frac{1}{n}\right) \\ \frac{1}{2}\left(\frac{4\|\rho\|_2^2}{kn} + \frac{\|\rho\|_2^2}{k} + \frac{1}{n}\right) & \left(\frac{2}{k^2} + \frac{1}{k}\right)\|\rho\|_2^2 \end{pmatrix}, b = \begin{pmatrix} \frac{1}{n} \\ \frac{\|\rho\|_2^2}{k} \end{pmatrix} \tag{7}
$$

Note that $A$ and $b$ depend on the problem data ($k$, $n$, and $\|\rho\|_2$), and that $A$ is a positive semi-definite matrix (as $k$ and $n$ are non-negative integers, and $\|\rho\|_2$ is between 0 and 1). In addition, we can express the constraints on the original parameters ($\beta \geq 0$ and $0 \leq \alpha \leq 1$) as a non-negativity constraint in the new parameters ($\theta \succeq 0$).

### B.2  KKT CONDITIONS

The optimal hyperparameters are defined (see main text) as the solution to the minimization problem:

$$
\begin{aligned} \underset{\theta}{\text{minimize}} \quad & \frac{1}{2}\|A\theta - b\|_2^2 \\ \text{subject to} \quad & \theta \succeq 0 \end{aligned} \tag{8}
$$

where $\theta = \begin{pmatrix} \alpha\beta \\ (1-\alpha)\beta \end{pmatrix}$ are the hyperparameters to optimize, and $A$ and $b$ are specified in eq. (7).

The Lagrangian for (8) is given by $L(\theta, \lambda) = \frac{1}{2}\|A\theta - b\|_2^2 - \lambda^T\theta$, and the corresponding dual problem is:

$$
\begin{aligned} \underset{\lambda}{\text{maximize}} \quad & \inf_\theta \frac{1}{2}\|A\theta - b\|_2^2 - \lambda^T\theta \\ \text{subject to} \quad & \lambda \succeq 0 \end{aligned} \tag{9}
$$

Since the primal is convex, we have strong duality and the Karush-Kuhn-Tucker (KKT) conditions guarantee primal and dual optimality. These conditions include primal and dual feasibility, that the gradient of the Lagrangian vanishes ($\nabla_\theta L(\theta, \lambda) = A\theta - b - \lambda = 0$), and complimentary slackness (which ensures that for each inequality constraint, either the constraint is satisfied or $\lambda = 0$).

Solving the condition on the gradient of the Langrangian for $\lambda$ yields that the lagrange multipliers $\lambda$ are simply the residual $\lambda = A\theta - b$. Complimentary slackness tells us that $\lambda_i\theta_i = 0$, for all $i$.

We are interested in when this constraint becomes tight. To solve for this, we note that there are two regimes where each of the two inequality constraints is tight (the blue and orange regions in Figure 3a). These occur for the solutions $\theta^{(1)} = \begin{pmatrix} 0 \\ \frac{k}{k+2} \end{pmatrix}$ (when the first inequality is tight) and $\theta^{(2)} = \begin{pmatrix} \frac{n}{n+2} \\ 0 \end{pmatrix}$ (when the second inequality is tight). To solve for the transition point, we solve for the point where the constraint is tight *and* the lagrange multiplier ($\lambda$) equals zero. We have two inequality constraints, and thus will have two solutions (which are the two solid curves in Figure 3a). Since the lagrange multiplier is the residual, these points occur when $\left( A\theta^{(1)} - b \right)_1 = \lambda_1 = 0$ and $\left( A\theta^{(2)} - b \right)_2 = \lambda_2 = 0$.

The first solution $\theta^{(1)} = \begin{pmatrix} 0 \\ \frac{k}{k+2} \end{pmatrix}$ yields the upper bound:

$$\left( A\theta^{(1)} \right)_1 - b_1 = 0$$

$$\frac{1}{2}\left( \frac{1}{n} + \frac{\|\rho\|_2^2}{k} + 4\frac{\|\rho\|_2^2}{kn} \right)\left( \frac{k}{k+2} \right) = \frac{1}{n}$$

$$\|\rho\|_2^2 \left( \frac{n+4}{n} \right) = \frac{k+4}{n}$$

$$\|\rho\|_2 = \sqrt{\frac{k+4}{n+4}}$$

And the second solution $\theta^{(2)} = \begin{pmatrix} \frac{n}{n+2} \\ 0 \end{pmatrix}$ yields the lower bound:

$$\left( A\theta^{(2)} \right)_2 - b_2 = 0$$

$$\frac{1}{2}\left( \frac{1}{n} + \frac{\|\rho\|_2^2}{k} + 4\frac{\|\rho\|_2^2}{kn} \right)\left( \frac{n}{n+2} \right) = \frac{\|\rho\|_2^2}{k}$$

$$k + n\|\rho\|_2^2 + 4\|\rho\|_2^2 = \|\rho\|_2^2(2n+4)$$

$$\|\rho\|_2 = \sqrt{\frac{k}{n}}$$

These are the equations for the lines separating the regimes of optimal hyperparameters in Figure 3.

## C  ALTERNATIVE MOTIVATION FOR OPTIMAL HYPERPARAMETERS

Choosing hyperparameters which most rapidly descend the simple quadratic loss in eq. (10) is equivalent to choosing hyperparameters which minimize the expected square error in the estimated gradient, as is done in §3.4. This provides further support for the method used to choose hyperparameters in the main text. Here we derive this equivalence.

Assume a loss function of the form

$$f(x) = \frac{1}{2}\|x\|_2^2, \tag{10}$$

and that updates are performed via gradient descent with learning rate 1,

$$x \leftarrow x - g.$$

The expected loss after a single training step is then

$$\mathbb{E}_g \left[ f\left( x - g \right) \right] = \frac{1}{2}\mathbb{E}_g \left[ \|x - g\|_2^2 \right]. \tag{11}$$

For this problem, the true gradient is simply $\nabla f(x) = x$. Substituting this into eq. (11), we find

$$\mathbb{E}_g \left[ f(x - g) \right] = \frac{1}{2}\mathbb{E}_g \left[ \|\nabla f(x) - g\|_2^2 \right].$$

Up to a multiplicative constant, this is exactly the expected square error between the descent direction $g$ and the gradient $\nabla f(x)$ used as the objective for choosing hyperparameters in §3.4.

## D    COMPUTATIONAL AND MEMORY COST

Here, we outline the computational and memory costs of Guided ES and compare them to standard (vanilla) evolutionary strategies and gradient descent. As elsewhere in the paper, we define the parameter dimension as $n$ and the number of pairs of function evaluations (for evolutionary strategies) as $P$. We denote the cost of computing the full loss as $F_0$, and (for Guided ES and gradient descent), we assume that at every iteration we compute a surrogate gradient which has cost $F_1$. Note that for standard training of neural networks with backpropogation, these quantities have similar cost ($F_1 \approx 2F_0$), however for some applications (such as unrolled optimization discussed in §4.2) these can be very different.

| Algorithm | Computational cost | Memory cost |
|---|:---:|:---:|
| Gradient descent | $F_1$ | $n$ |
| Vanilla evolutionary strategies | $2PF_0$ | $n$ |
| Guided evolutionary strategies | $F_1 + 2PF_0$ | $(k+1)n$ |

Table 1: Per-iteration compute and memory costs for gradient descent, standard (vanilla) evolutionary strategies, and the method proposed in this paper, guided evolutionary strategies. Here, $F_0$ is the cost of a function evaluation, $F_1$ is the cost of computing a surrogate gradient, $n$ is the parameter dimension, $k$ is the subspace dimension used for the guiding subspace, and $P$ is the number of pairs of function evaluations used for the evolutionary strategies algorithms.

## E    EXPERIMENTAL DETAILS

Below, we give detailed methods used for each of the experiments from §4. For each problem, we specify a desired loss function that we would like to minimize ($f(x)$), as well as specify the method for generating a surrogate or approximate gradient ($\nabla \tilde{f}(x)$).

### E.1    QUADRATIC FUNCTION WITH A BIASED GRADIENT

Our target problem is linear regression, $f(x) = \frac{1}{2M} \|Ax - b\|_2^2$, where $A$ is a random $M \times N$ matrix and $b$ is a random $M$-dimensional vector. The elements of $A$ and $b$ were drawn IID from a standard Normal distribution. We chose $N = 1000$ and $M = 2000$ for this problem. The surrogate gradient was generated by adding a random bias (drawn once at the beginning of optimization) and noise (resampled at every iteration) to the gradient. These quantities were scaled to have the same norm as the gradient. Thus, the surrogate gradient is given by: $\nabla \tilde{f}(x) = \nabla f(x) + (b + n) \|\nabla f(x)\|_2$, where $b$ and $n$ are unit norm random vectors that are fixed (bias) or resampled (noise) at every iteration.

The plots in Figure 1b show the loss suboptimality ($f(x) - f^*$), where $f^*$ is the minimum of $f(x)$ for a particular realization of the problem. The parameters were initialized to the zeros vector and optimized for 10,000 iterations. Figure 1b shows the mean and spread (std. error) over 10 random seeds. For each optimization algorithm, we performed a coarse grid search over the learning rate for each method, scanning 17 logarithmically spaced values over the range $(10^{-5}, 1)$. The learning rates chosen were: 5e-3 for gradient descent, 0.2 for guided and vanilla ES, and 1.0 for CMA-ES. For the two evolutionary strategies algorithms, we set the overall variance of the perturbations as $\sigma = 0.1$ and used $P = 1$ pair of samples per iteration. The subspace dimension for Guided ES was set to $k = 10$. The results were not sensitive to the choices for $\sigma$, $P$, or $k$.

### E.2    UNROLLED OPTIMIZATION

We define the target problem as the loss of a quadratic after running $T = 15$ steps of gradient descent. The quadratic has the same form as described above, $\frac{1}{2M} \|Ax - b\|_2^2$, but with $M = 20$ and $N = 10$. The learning rate for the optimizer was taken as the output of a multilayer perceptron (MLP), with three hidden layers containing 32 hidden units per layer and with rectified linear (ReLU) activations after each hidden layer. The inputs to the MLP were the 10 eigenvalues of the Hessian, $A^T A$, and the output was a single scalar that was passed through a softplus nonlinearity (to ensure a positive learning rate). Note that the optimal learning rate for this problem is $\frac{2M}{\lambda_{\min} + \lambda_{\max}}$, where $\lambda_{\min}$ and $\lambda_{\max}$ are the minimum and maximum eigenvalues of $A^T A$, respectively.

The surrogate gradients for this problem were generated by backpropagation through the optimization process, but by unrolling only $T = 1$ optimization steps (truncated backprop). Figure 4b shows the distance between the MLP predicted learning rate and the optimal learning rate $\left(\frac{2M}{\lambda_{\min} + \lambda_{\max}}\right)$, during the course of optimization of the MLP parameters. That is, Figure 4b shows the progress on the meta-optimization problems (optimizing the MLP to predict the learning rate) using the three different algorithms (SGD, vanilla ES, and guided ES).

As before, the mean and spread (std. error) over 10 random seeds are shown, and the learning rate for each of the three methods was chosen by a grid search over the range $(10^{-5}, 10)$. The learning rates chosen were 0.3 for gradient descent, 0.5 for guided ES, and 10 for vanilla ES. For the two evolutionary strategies algorithms, we set the variance of the perturbations to $\sigma = 0.01$ and used $P = 1$ pair of samples per iteration. The results were not sensitive to the choices for $\sigma$, $P$, or $k$.

### E.3    SYNTHESIZING GRADIENTS FOR A GUIDING SUBSPACE

Here, the target problem consisted of a mean squared error objective, $f(x) = \frac{1}{2}\|x - x^*\|_2^2$, where $x^*$ was random sampled from a uniform distribution between [-1, 1]. The surrogate gradient was defined as the gradient of a model, $M(x; \theta)$, with inputs $x$ and parameters $\theta$. We parameterize this model using a multilayered perceptron (MLP) with two 64-unit hidden layers and relu activations. The surrogate gradients were taken as the gradients of $M$ with respect to $x$: $\nabla \tilde{f}(x) = \nabla_x M(x; \theta)$.

The model was optimized online during optimization of $f$ by minimizing the mean squared error with the (true) function observations: $L_{\text{model}}(\theta) = \mathbb{E}_{x \sim D} \left[ f(x) - M(x; \theta) \right]^2$. The data $D$ used to train $M$ were randomly sampled in batches of size 512 from the most recent 8192 function evaluations encountered during optimization. This is equivalent to uniformly sampling from a replay buffer, a strategy commonly used in reinforcement learning. We performed one $\theta$ update per $x$ update with Adam with a learning rate of 1e-4.

The two evolutionary strategies algorithms inherently generate samples of the function during optimization. In order to make a fair comparison when optimizing with the Adam baseline, we similarly generated function evaluations for training the model $M$ by sampling points around the current iterate from the same distribution used in vanilla ES (Normal with $\sigma = 0.1$). This ensures that the amount and spread of training data for $M$ (in the replay buffer) when optimizing with Adam is similar to the data in the replay buffer when training with vanilla or guided ES.

Figure 5a shows the mean and spread (standard deviation) of the performance of the three algorithms over 10 random instances of the problem. We set $\sigma = 0.1$ and used $P = 1$ pair of samples per iteration. For Guided ES, we used a subspace dimension of $k = 1$. The results were not sensitive to the number of samples $P$, but did vary with $\sigma$, as this controls the spread of the data used to train $M$, thus we tuned $\sigma$ with a coarse grid search.

### E.4    AUTOENCODERS WITH DISCRETE LATENT VARIABLES

We trained a vector quantized variational autoencoder (VQ-VAE) as defined in van den Oord et al. (2017) on MNIST. Our encoder and decoder networks were both fully connected neural networks with 64 hidden units per layer and ReLU nonlinearities. For the vector quantization, we used a small codebook (twelve codebook vectors). The dimensionality of the codebook and latent variables was 16, and we used 10 latent variables. To train the encoder weights, van den Oord et al. (2017) proposed using a straight through estimator Bengio et al. (2013) to bypass the discretization in the vector quantizer. Here, we use this as the surrogate gradient passed to Guided ES. Since the gradients are correct (unbiased) for the decoder and embedding weights, we do not use Guided ES on those variables, instead using first-order methods (Adam) directly. For training with vanilla ES or Guided ES, we used $P = 10$ pairs of function evaluations per iteration to reduce variance (note that these can be done in parallel).

