# OpenReview forum: "Guided Evolutionary Strategies: Escaping the curse of dimensionality in random search"
_ICLR.cc/2019/Conference_

### Official Review · AnonReviewer3 · 2018-11-02
**Interesting idea, but not really new**

**Rating:** 6
**Confidence:** 4

**Review:**

The idea of this paper is to accelerate the OpenAI type evolution strategy by introducing
1. non-isotropic distribution, where the covariance matrix is of form I + UU^t; and
2. external information such as a surrogate gradient to determine U.
The experiments show promising results. I think it is the right direction to go, however at the same time, these ideas are not really new.

The first point is well studied in the context of evolution strategies in e.g., (Sun et al., arxiv 2011), (Loshchilov, Evolutionary Computation 2015), (Akimoto et al, GECCO 2016). They all have covariance matrix of form I + UU^t or a bit richer. There are mainly two advantages over full CMA-ES or NES: 1) computationally cheap, and 2) faster adaptation of the covariance matrix. The current paper does not adapt the covariance matrix and use an external information to guide the distribution. Therefore, it is different from the above work, but I suggest to compare Guided ES with these methods to see the effect of external information purely.

The second point, using external information to change the distribution shape, is also investigated in reference (Hansen, INRIA TechRep 2011), where external information such as a good point or a good directions (gradient) is injected in order to adapt the covariance matrix.

---

> ### Author Response · Authors · 2018-11-15
> **Thank you for your review. Comments below:**
>
> Thank you for your review. Comments below:
>
> Regarding the first point, we agree that comparisons against other adaptive ES methods (CMA-ES and NES) would be very useful, and are currently performing these comparisons. We will update the paper with these results. However, there is a fundamental limitation with CMA-ES and NES in that they are both purely black box (no gradient information) optimizers. This can be especially seen in the current Figure 1 in the paper, where CMA-ES is not able to take advantage of the initial external gradient information available to Guided ES or SGD, thus fails to make quick progress on the problem--it does begin to accelerate past standard ES as the covariance begins to adapt. NES will have the same issue. Again, we are running experiments to confirm this intuition.
>
> Regarding the second point, we carefully read through the Hansen 2011 reference again and could find no reference on using external gradient information to adapt the covariance matrix. As far as we can tell, Hansen 2011 focuses purely on adapting the covariance using information from iterates encountered during the optimization trajectory. To the best of our knowledge, our work is the first to propose incorporating surrogate” gradient information into an ES algorithm, as well as to analyze the bias and variance of the resulting gradient estimate.

---

> > ### Comment · AnonReviewer3 · 2018-11-30
> > **Regarding Hansen 2011**
> >
> > Thank you for your response.
> >
> > As the title of Hansen 2011 tells, it of course consider injecting external solutions or directions. On the first page of this paper, they say,
> > --
> > External or modified proposal solutions or directions can have a variety of sources.
> > • a gradient or Newton direction;
> > • an optimal solution of a surrogate model built from already evaluated solutions;
> > • the best-ever solution seen so far;
> > --
> >
> > The authors might misunderstand Hansen 2011 since the external solutions and directions (gradients) are transformed as a solution in the algorithm description.

---

> > > ### Author Response · Authors · 2018-11-30
> > > **Multiple Hansen 2011 references!**
> > >
> > > Our apologies--we were talking about different Hansen 2011 references. We thought the reviewer was referring to "The CMA Evolution Strategy: A Tutorial" (http://www.cmap.polytechnique.fr/~nikolaus.hansen/cmatutorial110628.pdf), but only today noticed this technical report "Injecting External Solutions Into CMA-ES" (https://hal.inria.fr/inria-00628254/document). We apologize for the confusion.
> > >
> > > This Hansen 2011 reference is indeed very relevant, we thank the reviewer for bringing it to our attention. After a quick read, the main difference with our work is that we inject external information into the covariance matrix from which perturbations are sampled, whereas Hansen 2011 injections solutions by replacing the samples themselves. We are reading this reference more carefully and will update our paper accordingly.

---

### Official Review · AnonReviewer1 · 2018-11-05
**Improve random search by building a subspace of the previous surrogate gradients for derivative-free optimization; good results but paper lacks clarity and is quite hard to follow.**

**Rating:** 4
**Confidence:** 3

**Review:**

Summary: The paper proposes a method to improve random search by building a subspace of the previous k surrogate gradients, mixing it with an isotropic Gaussian distribution to improve the search. Results reported shows are good compared to other approaches for learning weights of neural networks. However, paper lacks clarity and is quite hard to follow.

Quality: The paper presents a well-designed approach that is able to deal with optimization in high dimensionality space, by building a lower order surrogate model build upon the previous gradients computed with this surrogate model. The analysis appears to be correct and provide credential to the approach. Results reported are very good. However, testing is relatively limited to few cases. More experimental results on a good set of problems with several methods would have made the paper stronger and more convincing.

Clarity: The paper is hard to follow. Maybe because I am not completely familiar with the topic, but many elements presented lacks some context. The authors appear clearly to be knowledgeable of their topics, but lacks the capacity to provide all required background to follow their thoughts. The method could have been better illustrated, I found that Fig. 1a not enough to explain the method, while Fig. 1b and other training curves not useful to understand the approach. Some pseudo-code to illustrate the use of the proposed method might certainly help to improve clarity. Sec. 3.2 is not enough to understand well the approach.

Originality: The approach is allowing a nice trade-off between pure random search and guide search through a surrogate model over a subspace of limited dimensionality. This is in-line of some work on the use of ES for training neural networks, but I am not aware of other similar work although I am not super knowledgeable of the field.

Significance: The approach can have its impact for optimizing deep networks with no gradient, but more exhaustive experimental testing would be required.

Pros and cons:
+ Sound approach
+ Good theoretical support of the approach (bias-variance analysis)
+ Great results reported
+ Of importance for optimizing without gradients
- Presentation of the method lacking many details and not very clear
- Overall quality of the paper is subpar, tend to be very textual and hard to follow in several parts
- Experiments are not exhaustive and detailed. Loss plots are provided for some methods compared. Looks more like a preliminary validation.

I think that if the paper can be rewritten to be more tight, clearer in its presentation, with figures and pseudo-code to illustrate the method better, with more exhaustive testing, it can be really great. Current, the method appears to be great, but the writing quality of the paper is not yet there.

---

> ### Author Response · Authors · 2018-11-15
> **Thank you for your review. Comments below:**
>
> Thank you for your review. Comments below:
>
> Regarding clarity, we appreciate the reviewer’s comments and have added more context throughout the paper. We have added pseudocode in the main text, and moved more of the bias-variance derivation to the appendix (to make room for more exposition of the method in 3.2).
>
> In addition, we would appreciate it if the reviewer could elaborate on which aspects of the paper they thought could use more context. Specifically, if Fig1a is insufficient to explain the method, what additional information would the reviewer have appreciated?
>
> Regarding experiments, we are running more baseline comparisons against other adaptive evolutionary strategy methods (CMA-ES and natural evolutionary strategies, or NES). If the reviewer has other specific suggestions as to what would make the experiments more exhaustive, we would appreciate them.

---

### Official Review · AnonReviewer2 · 2018-11-05
**Good idea but its relation with a similar approach is overlooked and analysis is oversimplified**

**Rating:** 5
**Confidence:** 3

**Review:**

In this manuscript, the authors propose an approach that combines random search with the surrogate gradient information. To this end, the proposed method samples from the subspace of the surrogate gradients. This subspace is constructed by storing the previous surrogate gradients. After several assumptions, the authors also a give a discussion on variance-bias trade-off as well as a discussion on hyperparameter optimization. The manuscript ends with numerical experiments.

The proposed guided search seems similar to (stochastic) quasi-Newton methods. For instance the form in (2) is indeed a rank-one update of the gradient. What is authors take on this relationship?

The analysis assumes that the gradient exists. The proposed method is interesting when the gradients are not available. Therefore, it is not clear in what sense this analysis would apply to general functions.  The authors also assume that the second order Taylor expression is exact. Is this absolutely necessary? Would the analysis work when the function is approximated locally with its second order expansion?

I guess the equation in (2) is satisfied irrespective of the distribution of the \epsilon_i vectors. If I am right, then what is the role of the particular distribution used for sampling from the subspace of surrogate gradients?

The authors state "ES has seen a resurgence in popularity in recent years (Salimans et al., 2017; Mania et al., 2018)." Both cited papers are not published in any conference or journal. Is there some recent but published work to support the statement?

---

> ### Author Response · Authors · 2018-11-15
> **Thank you for the review. Responses below:**
>
> Thank you for the review. Responses below:
>
> “The proposed guided search seems similar to (stochastic) quasi-Newton methods. For instance the form in (2) is indeed a rank-one update of the gradient. What is authors take on this relationship?”
> Quasi-Newton methods assume access to first-order information about the objective, whereas our focus is on black-box optimization. Critically, we do not assume that the “surrogate” gradient information we are provided is reliable. Providing a way to robustly use this information when it is useful and to discard it when it is not is our primary contribution.
>
> Our update is indeed inspired by quasi-Newton methods, and we now note this in the maintext. In particular, as the author notes, we adapt the search covariance with a history of k past gradient estimates similar to how the approximate inverse Hessian is updated according to the past k gradient evaluations in L-BFGS. However, we are not trying to approximate the inverse Hessian. In our application, we are updating the covariance of the distribution used to perturb parameters.
>
> “The analysis assumes that the gradient exists. The proposed method is interesting when the gradients are not available. Therefore, it is not clear in what sense this analysis would apply to general functions.  The authors also assume that the second order Taylor expression is exact. Is this absolutely necessary? Would the analysis work when the function is approximated locally with its second order expansion?“
> These assumptions were made largely to simplify the presentation of the bias-variance analysis.  In the deep learning applications we focused on, the gradient of most loss functions exist, however, they may not be tractable (due to intractable integrals over nuisance variables) or may not be useful (e.g., the gradient of a hard thresholding function is 0). We dropped the higher order Taylor remainder to declutter the exposition, however, the analysis still holds (up to higher order error terms) when the function is locally approximated to 2nd order around each iterate.
>
> “I guess the equation in (2) is satisfied irrespective of the distribution of the \epsilon_i vectors. If I am right, then what is the role of the particular distribution used for sampling from the subspace of surrogate gradients?”
> Correct, the equation in (2) does not depend on the particular distribution for \epsilon. We chose a Gaussian distribution because it is: simple, easy to sample from, and has bounded variance. This is also consistent with previous work on evolutionary strategies (CMA-ES and NES). It would be interesting to, in future work, explore different choices for this distribution either analytically or empirically.
>
> “The authors state "ES has seen a resurgence in popularity in recent years (Salimans et al., 2017; Mania et al., 2018)." Both cited papers are not published in any conference or journal. Is there some recent but published work to support the statement?”
> We have added additional published citations [1, 2, 3, 4] all of which use evolutionary strategies in concert with neural networks.
>
> We would appreciate if the reviewer could expand on their justification for the given score, or consider revising it in the context of our responses here.
>
> [1] Cui et al. Evolutionary Stochastic Gradient Descent for Optimization of Deep Neural Networks (NIPS 2018)
> [2] Houthooft et. al. Evolved Policy Gradients (NIPS 2018)
> [3] Ha and Schmidhuber. World Models. (NIPS 2018)
> [4] Ha, David. Neuroevolution for deep reinforcement learning problems. (GECCO 2018)

---

### Meta-Review · Area_Chair1 · 2018-12-15
**Related work is overlooked and not compared with.**

**Confidence:** 5
**Recommendation:** Reject

**Metareview:**

This paper proposes a “guided” evolution strategy method where the past surrogate gradients are used to construct a covariance matrix from which future perturbations are sampled. The bias-variance tradeoff is analyzed and the method is applied to real-world examples.

The method is not entirely new, and discussion of related work as well as comparison with them is missing. The main contribution is in the analysis and application to real-world examples, and the paper should be rewritten focusing on these contributions, while discussing existing work on this topic thoroughly.

Due to these issue, I recommend to reject this paper.